# A Potential Anti-Tumor Herb Bred in a Tropical Fruit: Insight into the Chemical Components and Pharmacological Effects of Momordicae Semen

**DOI:** 10.3390/molecules24213949

**Published:** 2019-10-31

**Authors:** Xiao-Rong Xu, Chuan-Hong Luo, Bo Cao, Run-Chun Xu, Fang Wang, Xi-Chuan Wei, Ting Zhang, Li Han, Ding-Kun Zhang

**Affiliations:** 1State Key Laboratory of Characteristic Chinese Medicine Resources in Southwest China, College of Pharmacy, Chengdu University of Traditional Chinese Medicine, Chengdu 611137, China; 2018KS159@stu.cdutcm.edu.cn (X.-R.X.); lch1379034021@163.com (C.-H.L.); caobo22wave@163.com (B.C.); Xrrings@163.com (R.-C.X.); zxcder159@163.com (X.-C.W.); tinazhangting@163.com (T.Z.); 2Key Laboratory of Modern Preparation of Traditional Chinese Medicine (TCM), Ministry of Education, Jiangxi University of Traditional Chinese Medicine, Nanchang 330004, China; cat689apple@163.com

**Keywords:** Momordicae Semen, anti-tumor, pharmacological action, chemical constituents, mechanism

## Abstract

Gac fruit (*Momordica cochinchinensis* Spreng) is a popular tropical fruit in southeast Asia. What is amazing is that its seeds (Momordicae Semen) and arils are traditional herbs with anti-tumor activity, and have protected human health for more than 1000 years. In recent years, its anti-tumor activity has received extensive attention and research. This manuscript summarized the chemical composition of saponins, fatty acids, volatile constituents, proteins, peptides, and other components from Momordicae Semen (MSE). The effect and mechanism of MSE and its extract on breast cancer, gastric cancer, lung cancer, esophagus cancer, melanomas, and human cervical epithelial carcinoma were discussed. In addition, its antioxidant, anti-inflammatory, and other pharmacological effects were also analyzed. We hope that this review will provide new ideas for the treatment of cancer and other diseases, and become a reference for the further research into complementary and alternative medicine.

## 1. Introduction

Momordicae Semen (MSE) is the dry and mature seed bred in gac fruit*. Momordica cochinchinensis* is native to Southeast Asia, Vietnam, and mainly distributed in tropical regions such as Thailand, Laos, Cambodia, China, India, etc, [1]. Its name varies a lot in different countries, such as Fak Kao (Thailand), Mak Kao (Laos), and Bhat Ke rala (India) [2]. Its local name in Vietnam is red gac [3,4]. MSE is called Mubiezi in China, which was first published in the Kai bao Materia Medica in Song Dynasty and is mainly produced in Guangxi, Sichuan and Hubei province. The morphological and molecular diversity of 42 species of MSE from Australia; central, northern, and southern Vietnam; and Thailand were studied. The largest and most heavy MSE is from central Vietnam, and the lightest and smallest comes from Thailand [5]. 

Gac fruit is a tropical fruit that is used as a health food and traditional medicine in East and Southeast Asia. In Vietnam, the red aril that surrounds the seeds of mature gac fruit is usually consumed in the traditional recipe Vietnamese “Xoi Gac” [4]. The fruit, especially aril, is rich in carotenoids, β-carotene, and lycopene [6], which can be used for the treatment of infantile rickets, xeroma, and night blindness, according to the traditional Vietnamese documents [7,8]. In Thailand, immature gac fruits and shoots are taken as vegetables. In Guangxi province, southwest of China, people also have the custom of eating *Momordica cochinchinensis* seedlings, which are rich in vitamin C, vitamin B_2_, lycopene, beta carotene, and total carotenoids [9,10,11,12,13]. The anatomy of gac fruit from Guangxi, China is shown in Figure 1.

In China, MSE is commonly used in combination and can be used for the treatment of various diseases such as paronychia, hemorrhoids, and neurodermatitis [14]. Modern studies have shown that MSE has abundant antineoplastic activity. We aimed to review the research progress of MSE and summarize the pharmacological action and mechanisms of MSE. Prospects and development trends for the application and study of MSE are also described. 

## 2. Chemical Composition

A large number of studies have shown that MSE mainly contains saponins, fatty acids, volatile constituents, terpenoids, lignin, steroids, proteins, peptides, and other components. 

Oleanolic triterpenoid saponins with disaccharide chains are the main saponins in MSE. They mainly include saponins I, saponins II, gypsogenin 3-*O*-β-d-galactopyranosy (1→2)-[α-l-rhamnopyranosyl (1→3)]-β-d-glucuronopyranoside, and quillaic acid 3-*O*-β-d-galactopyranosyl (1→2)-[α-l-rhamnopyranosyl (1→3)]-β-d-glucuronopyranoside. The disaccharide saponins Ia-Ie, Ha-He, and saponins I and II also could be isolated from the roots of *Momordica cochinchinensis* [15]. Saponins I and saponins II are the most representative ones [16]. The structures of the main compounds are shown in Figure 2.

The aril of gac contains a high concentration of oil that is composed of several types of fatty acids. Similarly, fatty acids are also rich in the seeds [17]. MSE contained primarily stearic acid (60.5%), smaller amounts of linoleic (20%), oleic (9%), and palmitic acids (5%–6%), and trace amounts of arachidic, etc. [8]. Fourteen kinds of fatty acids were identified from MSE, accounting for 89.32% of the total fatty acid content, of which the unsaturated fatty acid content was 41.91% [18]. Gac aril contained 22% fatty acids by weight, composed of 32% oleic, 29% palmitic, and 28% linoleic acids. The fatty acids concentration was 101.98 mg·g^−1^ edible portion in gac pulp [19]. The unsaturated fatty acids of MSE have various effects on the body, such as adjusting blood fat, cholesterol, blood pressure, and preventing cancer [18]. The structures of the main compounds are shown in Figure 3. 

The volatile constituents of MSE mainly consist of alkanes, esters, alcohols, ketones, ethers, aldehydes, and organic acids [20]. The structures of the main compounds are shown in Figure 4.

Terpenoids are common chemical constituents in plants, and the terpenoids from MSE are shown in Figure 5. 

Lignin is widely found in nature and has abundant pharmacological activities. The lignin from MSE are shown in Figure 6.

Steroids are also important components in MSE, and their structures are shown in Figure 7.

The main proteins and peptides isolated from *Momordica cochinchinensis* (MCo) seeds are MCoCI, MCCTI-1 [21], MCoCC-1, MCoCC-2 [22], MCoTI-I to MCoTI-III [23], MCo-3 to MCo-6 [24], Cochinin B [25], and cochinchinin [26]. A chymotrypsin inhibitor 7514-Da named MCoCI, which belongs to the potato I inhibitor family, was isolated from the extract of MSE. It is the first chymotrypsin-specific serine protease inhibitor isolated from Cucurbitaceae [27]. MCCTI-1 and MCoTI-I to MCoTI-III are trypsin inhibitors, but no trypsin inhibitory effect was observed for MCoCC-1 within the 0.04–0.5 µM range [22]. MCoTI-I and MCoTI-II are head-to-tail cyclized peptides, whereas MCoTI-III is an acyclic peptide. MCoTI-I and MCoTI-II are extremely stable and can penetrate cells due to its small molecular weight. Thus, they are favorable choices as a framework for drug design applications [28]. MCo-3 to MCo-6 are not potent trypsin inhibitors, but can slightly inhibit cancer cell line MDA-MB-231. Cochinchinin is a ribosomal-inactivating protein with high cytotoxicity isolated from MSE [26]. 

The structures of some peptides and the other major compounds are shown in Figure 8, and the main compounds in MSE are shown in Table 1.

## 3. Pharmacological Effects

Anti-tumor activity is the most representative activity within the pharmacological activities of MSE. The anti-tumor effect may be mainly related to its pharmacological action of anti-oxidation, anti-inflammatory, anti-virus, and immune adjuvant.

### 3.1. Anti-Tumor Effect

MSE has been used in China for the treatment of oncological diseases for many years [36]. It has been proven that MSE can inhibit different tumors, which are listed in Table 2. 

#### 3.1.1. Breast Cancer

Breast cancer is the most common malignancy in women. It was also the second leading cause of cancer death among women after lung cancer [46]. Radiation therapies [47], targeted therapies [48,49], and endocrinotherapies [50] can be used for treating breast cancer, while TEC (taxotere, epirubicin, and cyclophosphamide) and CEF (cyclophosphamide, epirubicin, and fluorouracil) are the most common therapy method, but patients treated with TEC had a higher rate of neutropenia and leucopenia [51,52]. 

Ethanol extract of MSE can relieve the suffering caused by breast cancer. After 48 h of treatment with the extract, MDA-MB-231 cells proliferation was inhibited, cells apoptosis was induced, and the cell cycles were blocked in the dosage range of 0.1 to 0.4 mg·mL^−1^. Treated with MSE extract for 48 h, apoptotic evidences such as nuclear fragmentation, fluorescence, and typical apoptotic morphology could be found. Furthermore, the effects on MDA-MB-231 cells were in a dose-dependent manner. The mechanism induced apoptosis, and G2/M phase cell cycle arrest in MDA-MB-231 cells takes place to decrease the expression of the phosphatidylinositol 3-kinase/ protein kinase B (PI3K/Akt) pathway [40]. 

The ethyl acetate extract of MSE has been shown to have anti-tumor activity and was proved both in vivo and in vitro. Nude mouse xenotransplant models that were induced by injecting MDA-MB-231 cells subcutaneously into the right axilla of the mice proved that the extract could inhibit the growth of tumors. An experimental group was intraperitoneally injected with the ethyl acetate extract in the dosage of 50 mg·kg^−1^ (the 10% lethal dose) and 25 mg·kg^−1^ (the 5% lethal dose) once a day for 14 days. The results showed that the tumors in the group treated with the extract (25 or 50 mg·kg^−1^) were smaller and lighter [30]. A cell test also confirmed the anti-tumor activity of MSE. After 48 h of treatment with the extract, it showed a significant anti-proliferative effect on MDA-MB-231 cells in the dosage range of 8 to 250 μg·mL^−1^. Furthermore, the 50% inhibitory concentration (IC_50_) was 35.04 µg·mL^−1^. Its anti-tumor effect is attributed to inducing cell death, G2-phase arrest, apoptosis, and was mainly related to octadecanoic acid (**9**, Table 1, Figure 3), [(2*S*)-2-[(2*R*)-4-hexadecanoyloxy-3-hydroxy-5-oxo-2*H*-furan-2-yl]-2-hydroxyethyl] hexadecanoate (**100**, Table 1, Figure 8), (2*E*,4*E*)-nona-2,4-dienal (**97**, Table 1, Figure 8), and other components [30].

The water extract of MSE also has effects on breast cancer ZR-75-30 cell lines. It blocked the cells’ growth, inhibited the migration and invasion, and downregulated the activity of MMP-2 and MMP-9 in a concentration-dependent manner between 30 and 120 μg·mL^−1^. The *IC_50_* of the extract at 24 h, 48 h, and 72 h were 93.24 μg·mL^−1^, 34.04 μg·mL^−1^, and 53.43 μg·mL^−1^, respectively. The potential mechanism is attributed to reducing the expression of MMP-2 and MMP-9 [39]. 

Xiaojin Capsule, a Chinese patent medicine that contains MSE, is clinically used for the treatment of breast cancer. The anti-tumor effect is also supported by the results of basic experimental research. It was demonstrated by the zebrafish model, which was induced by transplanting fluorescently labeled human breast cancer cells (Michigan Cancer Foundation-7, MCF-7) into the yolk sac of zebrafish. Different concentrations (26.47 μg·mL^−1^, 88.24 μg·mL^−1^, and 264.72 μg·mL^−1^) were dissolved in the water of fish farming, and the tumor inhibition rate was calculated after 48 h of treatment. The growth inhibition rates were 31.66%, 54.35%, and 53.77%, respectively. When the concentration was 264.72 μg·mL^−1^, the inhibitory rate of zebrafish neovascularization was 21.74%, and the apoptosis induction rate was 28.92%. These results indicated that Xiaojin Capsule has the ability to inhibit breast cancer effectively in high concentrations [53]. The effect and mechanism of MSE on breast cancer are summarized in Figure 9.

#### 3.1.2. Melanomas

Cutaneous melanoma causes around 90% of deaths in skin cancer patients [54]. The incidence has risen steadily worldwide over the last decades [55]. The incidence is increasing annually, and is currently 19.5 for men and 20.9 for women per 100,000 populations (World Standard Population year 2000) [56].

The anti-tumor effect of MSE is closely related to its extraction solvents, and the active parts were the aqueous extraction and ethanol extraction after comparing the effects of six different extraction solvents on melanoma cell lines (MM418C1 and D24). It was found there was a large amount of trypsin inhibitors in aqueous extraction, which could reduce the viability of MM418C1 and D24 melanoma cells. The effects of aqueous extracts and ethanol extracts of MSE on the growth of four different tissue-derived tumor cells (melanoma B16-F1, lung cancer cell A549, breast cancer cell MDA-MB-231, and esophageal carcinoma cells TE-13) in vitro were compared, and the results showed that aqueous extract can only inhibit the growth of B16-F1 cells weakly, but for the growth of the other three tumor cells, an effect can hardly be observed. On the contrary, ethanol extract could significantly inhibit the growth of four kinds of tumor cells, and the inhibition of B16–F1 melanoma cells was the most obvious [57]. Although a rich amount of saponins and phenolic compounds was detected in the water-saturated butanol part, it has no effect on the anti-cancer potential [39].

The ethyl acetate extract of MSE exhibited the strongest anti-proliferative effects on B16–F1 cells among other MSE fractions (water or petroleum ether) at the concentration of 5–200 μg·mL^−1^. B16–F1 cell proliferation was inhibited in a dose- and time-dependent manner between 5 and 40 μg·mL^−1^. Treated with 20 μg·mL^−1^ extract, melanin concentration significantly increased after 24 h, 48 h, and 72 h. The extract inhibited the proliferation of melanoma cells through inducing cell differentiation, which is characterized by proliferation inhibition, dendrite-like outgrowth, increased melanogenesis production, and an enhanced activity of tyrosinase. Its anti-melanoma effect is mainly attributed to (*E*)-3-(4-hydroxyphenyl)prop-2-enal (CMSP; **101**, Table 1, Figure 8). The mechanism is to regulate the MAPK signaling pathway to inhibit the growth of melanoma B16–F1 cells and induce their differentiation [45]. 

It is reported that CMSP is the key component in the ethyl acetate extract of MSE. CMSP can inhibit B16–F1 cells in vivo and in vitro and has no harm to any organs within the therapeutic dose. [58]. A mouse melanoma model was used to evaluate the effects of CMSP on B16–F1 cells’ growth. The experimental group was intraperitoneally injected with 2 mg·kg^−1^ of CMSP every two days for three weeks, and the tumor volume was measured. On the 25^th^ day, the mice were sacrificed and the tumor and liver were removed. It was found that the level of tumor tyrosinase in mice treated with CMSP increased, while the levels of melanoma markers S-100B, MIA-A, and MMP-9 decreased. Furthermore, the volumes and weights of the tumors in the mice of the experimental group were markedly lower than those in control mice. With different concentrations of CMSP (10, 20, or 40 μM) for 24, 48, or 72 h, the B16–F1 cells’ proliferation was inhibited in a dose- and time-dependent manner, and dendrite-like cellular protrusions can be found. Meanwhile, the expression level of tyrosinase was higher after treatment, while the expression of the melanoma cell markers S-100B and MIA were lower compared with control cells. Colony formation, cell migration, and the metastasis of B16–F1 cells can be inhibited by administering 20 μM CMSP for 24 h [58]. It indicated that the mechanism by which CMSP affects the viability of melanoma B16–F1 cells is the induction of B16-F1 melanoma cells differentiation via the Ras homolog gene family, member A- mitogen-activated protein kinase- Mitogen-activated protein kinase- Mitogen-activated protein kinase (RhoA–MAPK)signaling pathway.

In addition, MCoCC-1 (**85**, Table 1) is another anti-tumor active peptide isolated from MSE. Cytotoxicity assays were conducted to examine the effect of the peptides against four human cell lines, including colorectal adenocarcinoma (HT29), lung carcinoma cells (A549), melanoma cell line (MM96L), and a noncancerous control neonatal foreskin fibroblast cell line (NFF). It exhibited nonhemolytic and cytotoxic effects against human melanoma cell line MM96L [22].

Ethyl acetate, ethanol, and water extracts of MSE can all inhibit melanoma, and the monomer that plays a role is relatively clear. There may be synergies between them. The ethyl acetate extract of MSE can inhibit melanoma in vivo and in vitro. Furthermore, the compounds related to the efficacy have been clarified, and it is expected to provide a new solution for the treatment of melanoma. The anti-melanoma effects of different solvents are shown in Figure 10. 

#### 3.1.3. Lung Cancer

Lung cancer is the most commonly diagnosed cancer worldwide. In 2012, 1.8 million new lung cancer cases were estimated to have occurred globally, accounting for almost 13% of all new cancer cases (excluding non-melanoma skin cancer) [59].

MSE water extract can block Hepatoma 22 (H22) in vivo and A549 lung cancer cells in vitro. The anti-tumor effect on H22 was studied by an H22 subcutaneously transplanted tumor mice model. The tumor inhibition rate was 52.63% by giving 0.1 g·100g ^−1^ extract through intragastric administration for 3 weeks. The effect on A549 was studied by vitro. It was found that the water extract could block the proliferation of A459 cells and the IC_50_ was about 13.7 mg·mL^−1^. It also blocked the growth of A549, which was inhibited in a dose-dependent manner from 2.5 to 20 mg·mL^−1^. The growth inhibition rate could reach 76.76% by applying 20 mg·mL^−1^ water extract to the A549 for 24 h. The morphology of A549 became smaller and rounded from the original rhombus with 5–20 mg·mL^−1^ water extract for 24 h [37]. 

The ethanol extract of MS plays an important role in the inhibition of lung cancer cells. The IC_50_ on lung cancer cells A549 and H1299 were 0.38 and 0.21 mg·mL^−1^ at 72 h, respectively. The extract dramatically inhibited the proliferation of A549 and H1299 from 0.05 to 0.4 mg·mL^−1^. The extract significantly induced apoptosis, and blocked the migratory and invasive performance of A549 cells between 0.1 and 0.4 mg·mL^−1^ for 48 h. The expression of STAT-3, MMP-2, and the level of B-cell lymphoma-2(Bcl-2) were significantly decreased, while the levels of E-cadherin, p53, and Bcl-2 Associated X Protein(Bax) were increased by administering the extract (0.2 and 0.4 mg·mL^−1^) for 48 h. The extract blocked the proliferation of A549 cells by inducing apoptosis, at least partly through the activation of p53 and inactivation of PI-3K/Akt signaling. The extract induces A549 apoptosis through the mitochondrial pathway and inhibits metastasis by the STAT-3 and MMP-2 pathways [41].

#### 3.1.4. Gastric Cancer

Gastric cancer is still the fourth most common malignancy and remains the third leading cause of cancer-related death, following lung cancer and liver cancer [59]. 

The ethanol extract of MSE could inhibit gastric cancer cells GC7901 and MKN-28 in vitro. The *IC_50_* values on SGC7901 and MKN-28 were 0.38 and 0.20 mg·mL^−1^, respectively. The extract induced the apoptosis of SGC7901 and MKN-28 and blocked the cell cycle at the S phase from 0.1 to 0.4 mg·mL^−1^ after treating for 48 h. It significantly downregulated the protein levels of Bcl-2 and PARP, but upregulated the protein levels of fas-associating protein with a novel death domain(Fas/FADD), Bax, and p53. Meanwhile, it has the potential to increase the activity of caspase-3 and caspase-9, and slightly affected that of caspase-8 in SGC-7901. The mechanism is via the regulation of the PARP and p53 signal pathways [42]. 

#### 3.1.5. Esophageal Cancer

Esophageal cancer is the eighth most common cancer worldwide and the sixth leading cause of cancer-related mortality [59]. It is often accompanied with a poor prognosis, and the five-year survival ranges from 4% to 40% [60].

CMSP isolated from the ethanol extract is a major chemical component that inhibits some esophageal cancer cells. It was found that it inhibited the proliferation of Kyse30, TE-13, Ecal09, and Kyse180 at the concentration of 20 μg·mL^−1^. The cell cycles of TE-13 and Kyse30 could be blocked at the G0/G1 phase, and typical branched cell processes could be found. After 24 h, 48 h, and 72 h of administration, the expression levels of carcino-embryonic antigen (CEA) and squamous cell carcinoma antigen (SCC) were obviously lower than those in the control group from 10 to 40 μg·mL^−1^. Its mechanism is related to the upregulation of p-P38 in the MAPK signaling pathway and the downregulation of phosphorylation extracellular signal-regulated kinase (p-ERK) and phosphorylation Serine/threonine protein kinase/c-Jun /N-terminal kinase (p-SPKA/JNK) 43]. It can also restrain the migration and invasion of Kyse30 cells. Simultaneously, CMSP could decrease the expression levels of C-myc and N-myc in Kyse30 cells dramatically. It works by activating the RhoA–MAPK pathway and inducing the differentiation of Kyse30 cells [61]. 

#### 3.1.6. Human Cervical Epithelial Carcinoma

Cervical cancer is the second most common cancer in females worldwide. Most are squamous cell carcinomas. Adenocarcinoma of the uterine cervix accounts worldwide for 10%–20% of all cervical cancers [62].

Cochinin B is a ribosome-inactivating protein that originates from MSE. It is a single-chain protein with a molecular weight of 28 kDa [25]. It is considered to have potent and broad anti-tumor activity. Cochinin B is very effective against human cervical epithelial carcinoma (HeLa), human embryonic kidney (HEK293), and human small cell lung cancer (NCI-H187) cell lines with *IC_50_* values of 16.9, 114 and 574 nM, respectively [25]. 

### 3.2. General Pharmacological Action

In addition to antineoplastic activity, the other pharmacological effects of MSE are listed in Table 3.

#### 3.2.1. Anti-Inflammatory

The anti-inflammatory effect of MSE is mainly attributed to the fatty oil and saponin components, and the content of fatty oil affects its anti-inflammatory effect, which was confirmed by an ear swelling model in mice. It was found that when the oil content is 20%, the anti-inflammatory and analgesic effects of MSE were most obvious [71]. Saponins also account for the anti-inflammatory effect. Quillaic acid 3-*O*-β-d-galactopyranosyl(1→2)-[α-L-rhamnopyranosyl(1→3)]-β-D-glucuronopyranoside (**4**, Table 1) showed an anti-inflammatory activity in RAW 264.7 cells by reducing the production of nitric oxide (NO) at the concentration of 20 µM for 1 h. The mechanism is to block the expression of nitric oxide and IL-6 via the NF-κB pathway, which was induced by lipopolysaccharides [29]. In addition, saponins I (**1**, Table 1) have the potential to treat inflammatory diseases and modulate macrophage activation. It can decrease NO production and downregulate the mRNA level of inducible NO synthase (iNOS) and cyclooxygenase (COX)-2 in lipopolysaccharide (LPS)-activated RAW 264.7 cells from 20 to 40 µM. It can also control the translocation of p65 and p50 (the subunit of the transcription factor NF-κB) to the nucleus. The phosphorylation levels of inflammatory signaling proteins (IκBα, Src, and Syk) are reduced [63]. 

#### 3.2.2. Antioxidant Activity

Both gac fruit and its seeds have antioxidant activity. In the fruit, vitamin E is the main natural antioxidant component [12]. Naturally occurring chymotrypsin inhibitors can prevent H_2_O_2_ formation [72]. Similarly, the chymotrypsin inhibitor MCoCI is the main antioxidant active ingredient in MSE [73]. The primary rat hepatocyte system showed that MCoCI pretreating hepatocytes for 24 h can significantly reverse the cell damage induced by tert-butyl hydroperoxide (t-BHP), as well as associated glutathione depletion and lipid peroxidation at 100 µg·mL^−1^ [74]. MCoCI can protect cells from acute oxidative stress induced by exogenous drugs to some extent, and its mechanism includes the prevention of reactive oxygen species formation [64]. In addition, the total antioxidant capacity (T-AOC), hydroxyl radical scavenging ability, and catalase activity of the extracts from the alkaloids part of MSE were determined by kit experiments, and the extract of MSE concentration was found to be positively correlated with T-AOC in the range of 26.880–94.080 μg·mL^−1^ [75]. Various reactive oxygen species in the body could cause oxidative damage to biological macromolecules such as nucleic acids (DNA, RNA), lipids, and proteins in the body, which leads to the occurrence of disease [76]. 

#### 3.2.3. Immune Adjuvant

The ethanol extract of MSE is a relatively safe immune adjuvant. The ethanol extract of MSE could be used as a supplementary treatment for Newcastle disease (ND). Forty-eight chickens were immunized with the ND vaccine mixed with 0, 20, 40, or 80 μg of the extract for 35 days. It was found that the humoral immune response was enhanced and the vaccine mixed with 80 μg extract had the best effect. [77]. MSE has been shown to be effective in enhancing the immune response of chickens to avian influenza (H5N1) vaccination [78]. The FMDV (Foot-and-mouth disease virus) vaccine is a commercial oil-adjuvanted bivalent vaccine (serotypes O and Asia I). However, the probability of an effective immune response caused by vaccination for FMDV is small [79,80,81]. The immune adjuvant effect of MSE on the FMDV vaccine was also confirmed. It was found that the humoral immune response was increase by the oral administration or subcutaneous injection of the extract after inoculation with the FMDV vaccine. The extract tended to enhance serum immunoglobulin G (IgG) and IgG isotypes of mice immunosuppressed by a subcutaneous injection of dexamethasone [65]. Meanwhile, it has an adjuvant effect on the immune response induced by ovalbumin and has a low hemolytic activity [82]. Ovalbumin and MSE extract act in mice to induce higher specific antibody production, and ovalbumin-specific IgG titers are significantly increased among 50 to 100 μg [82]. In addition, MCoCI has immunopotentiating and anti-inflammatory effects [83]. MCoCI also can stimulate the proliferation of spleen cells, spleen lymphocytes, and bone marrow cells in the immune system, and promote the mitosis of spleen cells, which is usually accompanied by the induction of cytokines, such as IL-2 [84]. Quillaja saponaria Molina (Quil A) and aluminum salts used to be effective immune adjuvants, but their use was limited [85]. Quil A has hemolysis activity, and aluminum salts were found to be associated with muscle disease in humans [86]. By contrast, MSE is more safe and effective. 

#### 3.2.4. Antibacterial

The antibacterial and acaricidal effects are due to the saponins. It was found that the water extract of MSE can block the growth of *Candida albicans*, *Staphylococcus aureus*, *Escherichia coli*, and *Pseudomonas aeruginosa* [66]. 

#### 3.2.5. Protect Kidney Cell

It was discovered that the protective effect of the MSE on cisplatin-induced renal cell injury was mainly attributed to gypsogenin 3-*O*-β-d-galactopyranosy (1→2)-[α-l-rhamnopyranosyl (1→3)]-β-d-glucuronopyranoside (**3**, Table 1) and quillaic acid 3-*O*-β-d-galactopyranosyl (1→2)-[α-L- -rhamnopyranosyl (1→3)]-β-d-glucuronopyranoside (**4**, Table 1, Figure 2). They can decrease the toxicity of LLC-PK1 pig kidney epithelial cells induced by cisplatin at both 5 and 25 μM. Its mechanism is related to blocking the MAPK signaling cascade [67].

#### 3.2.6. Promote the Healing of Gastric Ulcer 

The ethanol extract of MSE can accelerate the ulcer healing of a rat model that is induced by ethanol and water immersion restraint stress (WRS). The extract was administrated intragastrically in a volume of 5 mL·kg^−1^ 1 h before the WRS. It decreased gastric damage at the dosage of 50 to 100 mg·kg^−1^. The extract resists acute gastric mucosal damage by suppressing the production of pro-inflammatory cytokines, downregulating cytosolic phospholipase A_2_ (cPLA2) and 5-lipoxygenase (5-LOX), and increasing mucus synthesis [87]. Similarly, it can also accelerate the ulcer healing of another model of gastric ulcer induced by acetic acid. Administered orally at a dose of 200 mg·kg^−1^ extract once per day for 14 days after the acetic acid injection, the expression of mRNA (at day 7) as well as vascular endothelial growth factor (VEGF) protein (at day 14) were significantly ascended. Its mechanism is related to the upregulation of VEGF and angiogenesis in a rat model [67]. 

### 3.3. Induce Neurite Outgrowth

MSE have a nutritional effect on the nerves. Neurite length could reach to 40.20 ± 2.72 µm (seed extract treatment of 138 µg·mL^−1^) and 3.58 ± 0.42 µm (controls). The seed extract presented the same behavior with nerve growth factor (NGF) as the neurotrophic effects [17]. The 17-kDa protein obtained from MSE has the potential to induce neurite outgrowth, which was proved by a PC-12 cell model [69]. Nerve growth factor (NGF) has the ability to induce neuronal differentiation and repair the central nervous system. However, its use is limited because it fails to pass through the blood–brain barrier. 

### 3.4. Prevented the Damages of Reproductive 

MSE aril extract can prevent adverse male reproductive parameters and testicular damage induced with valproic acid (VPA). Male Wistar rats were pretreated with aqueous extract for 23 days before being co-administered with VPA induction for 10 days. It was found that MSE extract in various dosages (50, 100, and 200 mg·kg^−1^) could significantly improve the relative weights of epididymis plus vas deferens in VPA rats. The extract can not only prevent the damages of testicular tissue and seminiferous tubule diameters, it can also be safe for use. Its high antioxidant activities may be attributed to the mechanism of preventing testicular damage [70]. 

## 4. Toxicity

Although it is a seed of tropical fruit and has a medicinal history of nearly 1000 years in China, it is not a completely non-toxic herb as desired. The famous Compendium of Materia Medica in the Ming Dynasty recorded its toxicity. However, the exposure of toxicity in clinical application is rare. The reasons are as follows: firstly, the main way of administration is transdermal administration. Secondly, when taking medicine orally, it is necessary to remove fatty oils, and the daily dose is between 0.6 and 1.2 g [88]. 

Modern studies have shown that the toxic components in MSE are mainly saponin and cochinchinin. Both the water extract and the ethanol extract of MSE have certain toxicity [89,90,91,92]. Studies have shown that the median lethal dose (LD_50_) of intravenous injection of saponin in mice was 32.35 mg·kg^−1^, 37.34 mg·kg^−1^ in intraperitoneal injection, and 0.9582 g·kg^−1^ in oral. Furthermore, saponin has a hemolysis effect on rabbit red blood cells [89]. Another toxic component is cochinchinin. The mouse has an intraperitoneal injection of LD_50_ of 16 mg·kg^−1^, and the poisoned mouse died quietly [89]. The toxicity of the aqueous extract of MSE was assessed by the oral and intraperitoneal injection of 0.5 ml ·100 g^−1^ in mice for seven days. It was found that the LD_50_ of the extract was 4.03 g·kg^−1^ by oral administration and 146.17 mg·kg^−1^ by intraperitoneal injection [90]. The toxicity of high-concentration MSE water extract was evaluated by clinical trials. After approximately 8 to 12 g of MSE water extract is applied to 50 tumor patients, alanine aminotransferase is elevated, and may cause mild damage to the liver after long-term use. However, it is basically safe [92]. The toxicity of CMSP from MSE ethanol extract was observed by intraperitoneally injecting CMSP with 400 mg·kg^−1^ 3 times per week for 28 days in mice. The change in the rate of micronucleus formation in mouse bone marrow cells was not obvious, and pathologic changes in visceral tissues can hardly be seen. CMSP has a relatively small toxic effect within the effective dosage range [91]. 

Many patients with lung cancer still suffer from facial and neck edema in the middle and late stages; five cases of pulmonary cancer edema were treated with MSE as the main drug. The results showed that the symptoms of edema disappeared, all kinds of symptoms were relieved, the condition was stable, and there were no obvious adverse reactions. Furthermore, the quality of life improved sharply [93]. 

## 5. Outlook

The anti-tumor activity of MSE is worthy of further investigation and development. In traditional medication experiences, it is often crushed into powders or pills are prepared after removing excessive fatty oils [94]. In fact, most of the chemical constituents may be involved in the anti-tumor effect. Modern research also shows that various extraction parts all play a role in blocking cancer cell growth, reproduction, migration, activation, or inhibition. Both traditional and modern evidences indicate that its anti-tumor effect may be a combination of multiple components.

The extraction solvent has a considerable influence on the pharmacological activity of MSE. Numerous studies have shown that different solvents MSE extracts have effects on different cancers, or different cancer cell lines that act on the same cancer. The combination of extracts from different solvents may work synergistically, such as breast cancer and melanoma. In addition, extracts from the same solvent can also block different cancers. For example, ethanol extract can inhibit both lung cancer and breast cancer. In order to achieve the desired therapeutic effect, in the clinical application, the extract of different solvents should be reasonably selected. Moreover, the purification and identification of compounds in different solvents is important for the clinical application and discovery of new treatment options.

It is of importance to fully explore the clinical use value of MSE. Although basic research studies have reinforced the anti-tumor activity of MSE, it is necessary to conduct clinical observation on the efficacy of MSE, and to systematically evaluate the anti-tumor adaptation range, optimal indication, and dose–effect relationship of MSE. Except for the inhibition of cancer, the improvement of clinical symptoms and the quality of life of patients should also be assessed due to the anti-inflammatory and analgesic activity of MSE. 

At present, there are limited studies on the anti-tumor effects of MSE. Furthermore, relevant clinical trials have not yet started or been published until now. MSE deserves additional attention in the anti-tumor research field. 

## Figures and Tables

**Figure 1 molecules-24-03949-f001:**
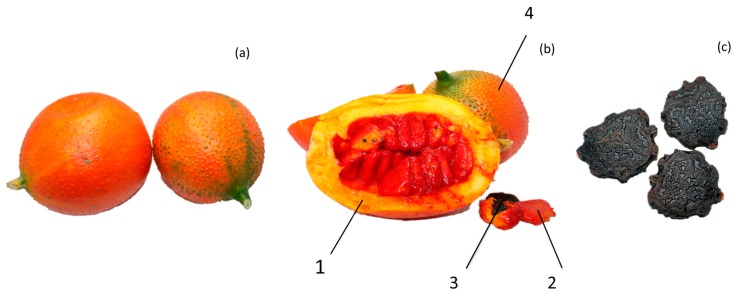
The anatomy of gac fruit: (**a**) gac fruit, (**b**) longitudinal section of fruit (1. Pulp, 2. Aril, 3. Seed, 4. Peel with spines), (**c**) the seeds of gac frui.

**Figure 2 molecules-24-03949-f002:**
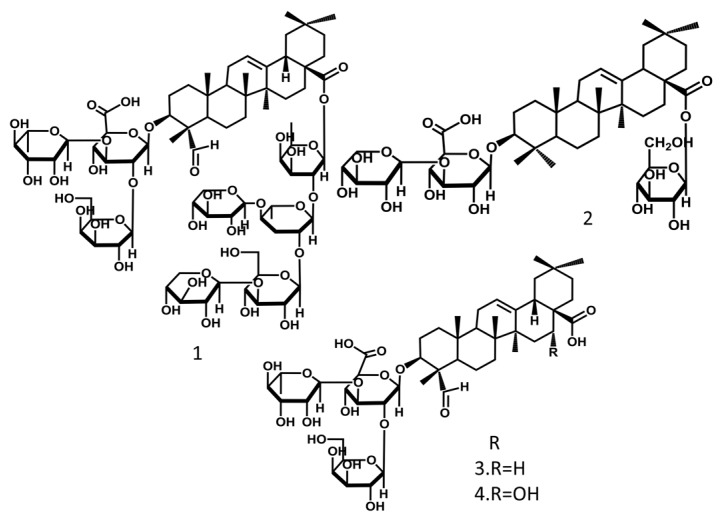
Saponins from Momordicae Semen.

**Figure 3 molecules-24-03949-f003:**
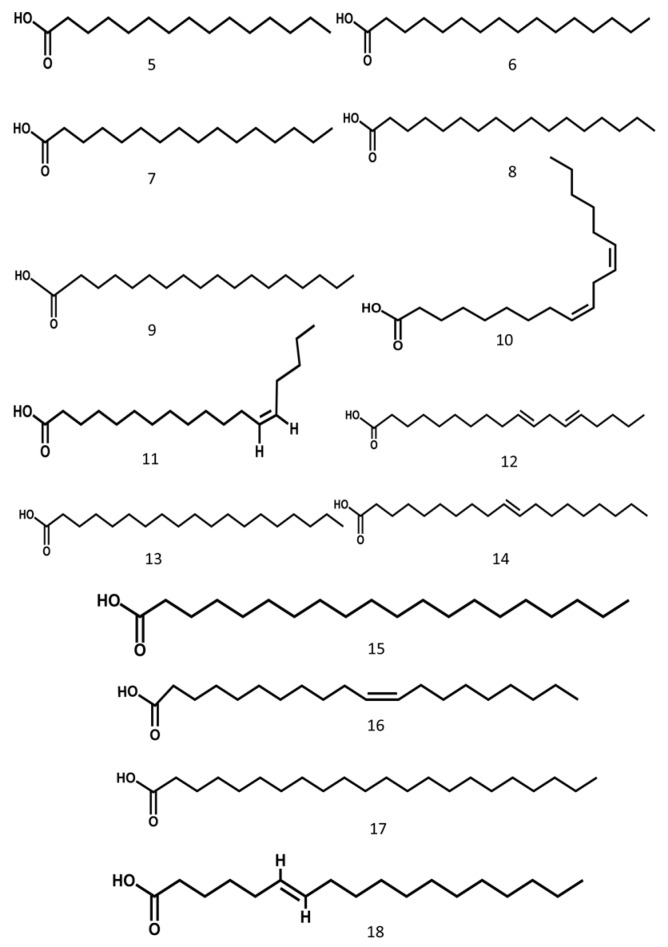
Fatty acid from Momordicae Semen.

**Figure 4 molecules-24-03949-f004:**
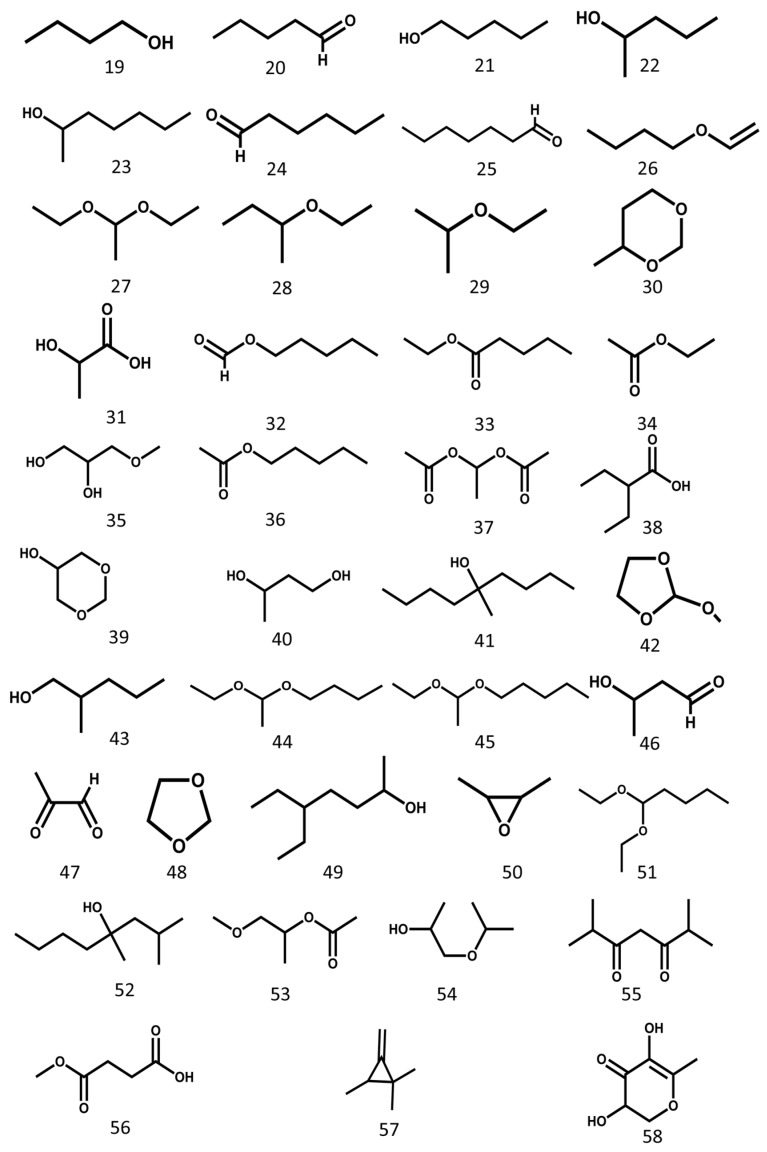
Volatile constituents from Momordicae Semen.

**Figure 5 molecules-24-03949-f005:**
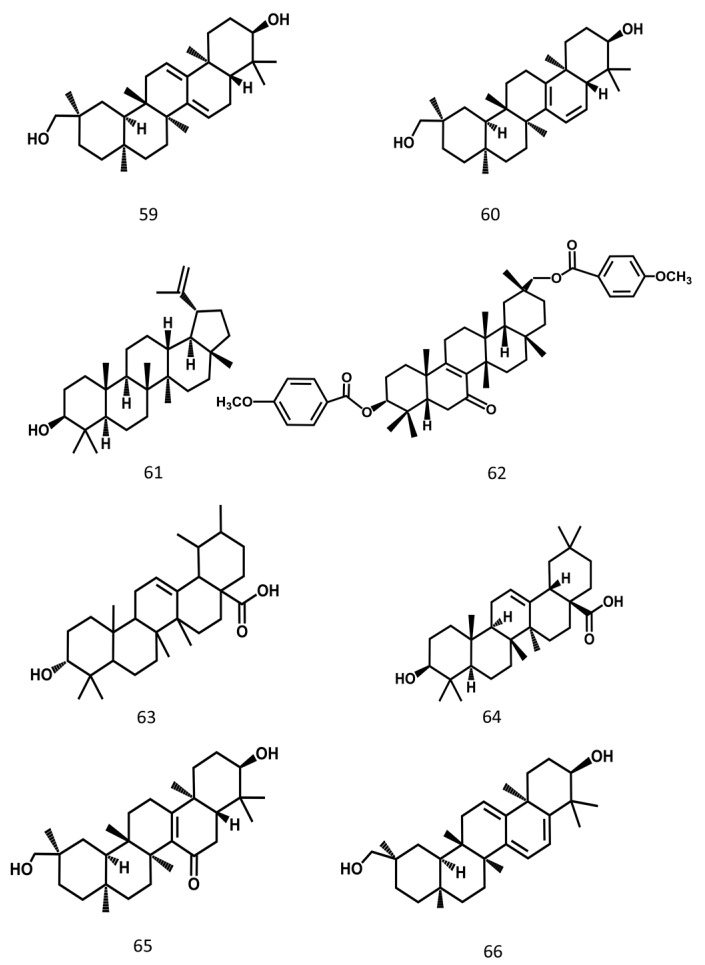
Terpenoids constituents from Momordicae Semen.

**Figure 6 molecules-24-03949-f006:**
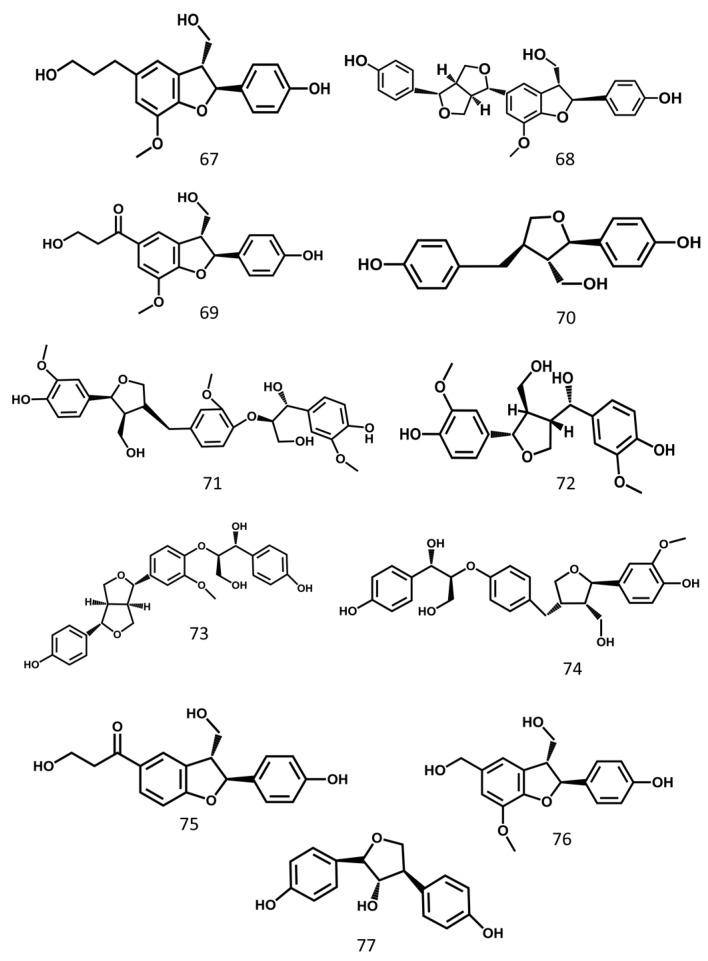
Lignin from Momordicae Semen.

**Figure 7 molecules-24-03949-f007:**
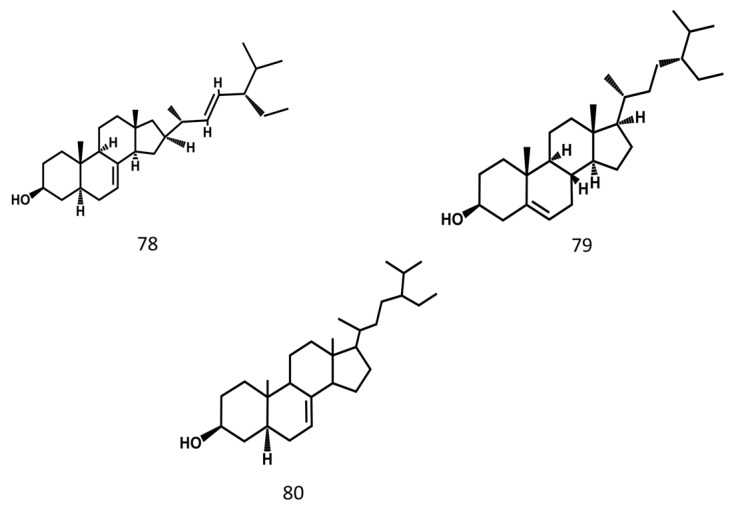
Steroids from Momordicae Semen.

**Figure 8 molecules-24-03949-f008:**
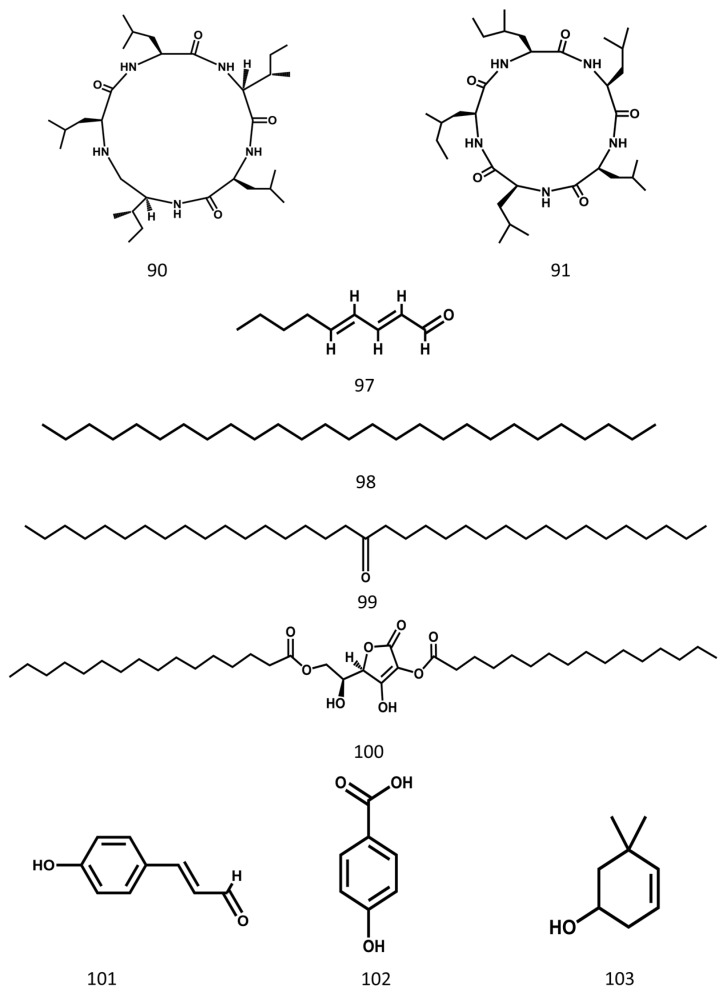
Some peptides and other constituents from Momordicae Semen.

**Figure 9 molecules-24-03949-f009:**
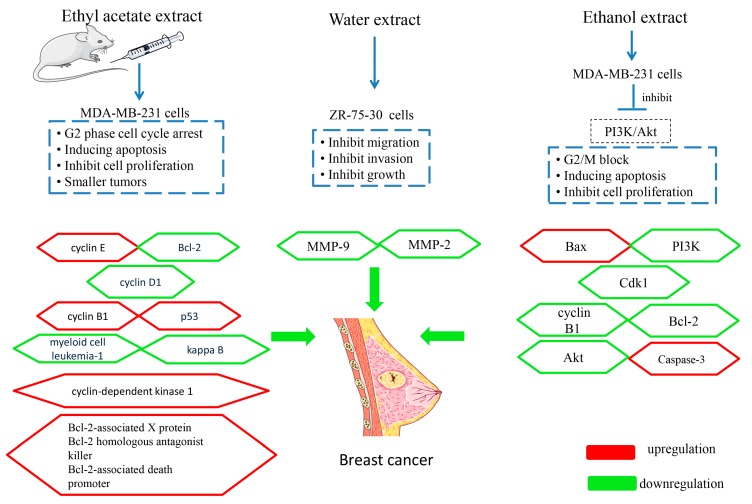
Anti-breast cancer effects and mechanisms of Momordicae Semen.

**Figure 10 molecules-24-03949-f010:**
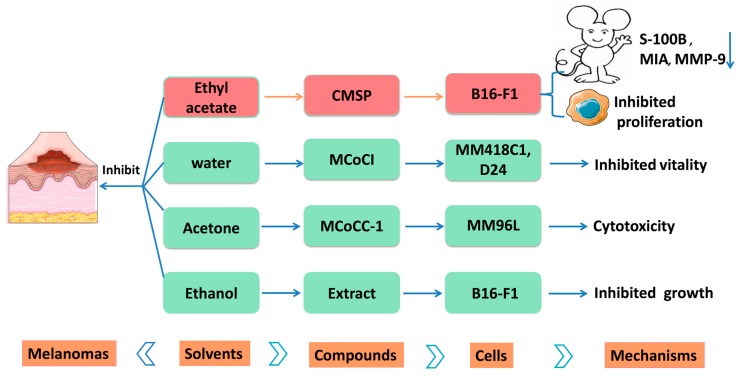
Anti-melanoma effects and mechanisms of Momordicae Semen.

**Table 1 molecules-24-03949-t001:** Main chemical components in Momordicae Semen.

No.	Chemical Class	Specific Chemical Composition	Refs
1	**Saponins**	Saponins I	[3]
2		Saponins II	[3]
3		Gypsogenin 3-O-β-D-galactopyranosy (1→2)- [α-L-rhamnopyranosyl (1→3)]-β-D-glucuronopyranoside	[29]
4		Quillaic acid 3-O-β-D-galactopyranosyl (1→2)-[α-L- -rhamnopyranosyl (1→3)]-β-D-glucuronopyranoside	[29]
5	**Fatty acids**	Pentadecanoic acid	[18]
6		Hexadec-11-enoic acid	[18]
7		Hexadecanoic acid	[18]
8		Heptadecanoic acid	[18]
9		Octadecanoic acid	[18]
10		(9Z,12Z)-octadeca-9,12-dienoic acid	[18]
11		(Z)-octadec-13-enoic acid	[18]
12		Octadeca-10,13-dienoic acid	[18]
13		Nonadecanoic acid	[18]
14		Nonadec-10-enoic acid	[18]
15		Icosanoic acid	[18]
16		(Z)-icos-11-enoic acid	[18]
17		Docosanoic acid	[30]
18		Octadec-6-enoic acid	[31]
		4-hydroxybenzoic acid	
19		Butan-1-ol	[20]
20	**Volatile** **constituents**	Pentanal	[20]
21		Pentan-1-ol	[20]
22		Pentan-2-ol	[20]
23		Heptan-2-ol	[20]
24		Hexanal	[20]
25		Heptanal	[20]
26		1-ethenoxybutane	[20]
27		1,1-diethoxyethane	[20]
28		2-ethoxybutane	[20]
29		2-ethoxypropane	[20]
30		4-methyl-1,3-dioxane	[20]
31		2-hydroxypropanoic acid	[20]
32		Pentyl formate	[20]
33		Ethyl pentanoate	[20]
34		Ethyl acetate	[20]
35		3-methoxypropane-1,2-diol	[20]
36		Pentyl acetate	[20]
37		1-acetyloxyethyl acetate	[20]
38		2-ethylbutanoic acid	[20]
39		1,3-dioxan-5-ol	[20]
40		Butane-1,3-diol	[20]
41		5-methylnonan-5-ol	[20]
42		2-methoxy-1,3-dioxolane	[20]
43		2-methylpentan-1-ol	[20]
44		1-(1-ethoxyethoxy)butane	[20]
45		1-(1-ethoxyethoxy)pentane	[20]
46		3-hydroxybutanal	[20]
47		2-oxopropanal	[20]
48		1,3-dioxolane	[20]
49		5-ethylheptan-2-ol	[20]
50		2,3-dimethyloxirane	[20]
51		1,1-diethoxypentane	[20]
52		2,4-dimethyloctan-4-ol	[20]
53		1-methoxypropan-2-yl acetate	[20]
54		1-(1-methylethoxy)-2-propanol	[20]
55		2,6-dimethylheptane-3,5-dione	[20]
56		4-methoxy-4-oxobutanoic acid	[20]
57		1,1,2-trimethyl-3-methylenecyclopropane	[20]
58		3,5-dihydroxy-6-methyl-2,3-dihydropyran-4-one	[20]
		Trimethylsilyl	
59	**Terpenoid**	(3R,4aR,6aS,6bS,8aS,11R,12aR,14bS)-11-(hydroxymethyl)-4,4,6a,6b,8a,11,14b-heptamethyl-1,2,3,4a,5,7,8,9,10,12,12a,13-dodecahydropicen-3-ol	[31]
60		(3R,4aR,6aS,6bS,8aS,11R,12aR,14bS)-11-(hydroxymethyl)-4,4,6a,6b,8a,11,14b-heptamethyl-1,2,3,4a,7,8,9,10,12,12a,13,14-dodecahydropicen-3-ol	[31]
61		(1R,3aR,5aR,5bR,7aR,9S,11aR,11bR,13aR,13bR)-3a,5a,5b,8,8,11a-hexamethyl-1-prop-1-en-2-yl-1,2,3,4,5,6,7,7a,9,10,11,11b,12,13,13a,13b-hexadecahydrocyclopenta[a]chrysen-9-ol	[32]
62		3,29-di-o-(p-methoxy)benzoylmultiflora-8-ene- 3 a,29-diol-7-one	[33]
63		(10R)-10-hydroxy-1,2,6a,6b,9,9,12a-heptamethyl-2,3,4,5,6,6a,7,8,8a,10,11,12,13,14b-tetradecahydro-1H-picene-4a-carboxylic acid	[34]
64		(4aS,6aR,6aS,6bR,8aR,10S,12aR,14bS)-10-hydroxy-2,2,6a,6b,9,9,12a-heptamethyl-1,3,4,5,6,6a,7,8,8a,10,11,12,13,14b-tetradecahydropicene-4a-carboxylic acid	[34]
65		(3R,4aR,6aS,6bS,8aS,11R,12aR,14bS)-3-hydroxy-11-(hydroxymethyl)-4,4,6a,6b,8a,11,14b-heptamethyl-2,3,4a,5,7,8,9,10,12,12a,13,14-dodecahydro-1H-picen-6-one	[31]
66		(3R,6aS,6bS,8aS,11R,12aR,14bR)-11-(hydroxymethyl)-4,4,6a,6b,8a,11,14b-heptamethyl-1,2,3,7,8,9,10,12,12a,13-decahydropicen-3-ol	[31]
67	Lignin	4-[(2S,3R)-3-(hydroxymethyl)-5-(3-hydroxypropyl)-7-methoxy-2,3-dihydro-1-benzofuran-2-yl]phenol	[32]
68		7-methoxy-2,3-dihydro-1-benzofuran-5-yl]-1,3,3a,4,6,6a-hexahydrofuro[3,4-c]furan-3-yl]phenol	[32]
69		3-hydroxy-1-[(2S,3R)-3-(hydroxymethyl)-2-(4-hydroxyphenyl)-7-methoxy-2,3-dihydro-1-benzofuran-5-yl]propan-1-one	[32]
70		(7R, 8S, 8′R)-4, 4′, 9-trihydroxy-7, 9′-epoxy-8, 8′-lignan	[32]
71		(1S,2R)-1-(4-hydroxy-3-methoxyphenyl)-2-[4-[[(3R,4R,5S)-5-(4-hydroxy-3-methoxyphenyl)-4-(hydroxymethyl)oxolan-3-yl]methyl]-2-methoxyphenoxy]propane-1,3-diol	[32]
72		(7 R, 8 S, 7′ R, 8′ S, 7″ S, 8″ R)-4′-guaiacylglyceryl-3-demethoxypinoresinol	[35]
73		3-demethoxyl- ethletianol C	[35]
74		(7′R,8′S)-7-oxo-9, 4′, 9′-trihydroxy-4, 7′-epoxy -3, 8′-neolignan	[35]
75		(7′S,8′S)-4-methoxyl-7-hydroxylmethyl-9′-hydroxy-4, 7′-epoxy-3, 8′-neolignan	[35]
76		(2R, 3S, 4S)-2,4-bis-(4-hydroxy-phenyl)-tetrahydrofuran-3-ol	[35]
77		4-[(2S,3R,4S)-4-[(S)-hydroxy-(4-hydroxy-3-methoxyphenyl)methyl]-3-(hydroxymethyl)oxolan-2-yl]-2-methoxyphenol	[31]
78		(3S,5S,9R,10S,13R,14R,17R)-17-[(E,2R,5S)-5-ethyl-6-methylhept-3-en-2-yl]-10,13-dimethyl-2,3,4,5,6,9,11,12,14,15,16,17-dodecahydro-1H-cyclopenta[a]phenanthren-3-ol	[32]
79	Steroid	(3S,8S,9S,10R,13R,14S,17R)-17-[(2R,5R)-5-ethyl-6-methylheptan-2-yl]-10,13-dimethyl-2,3,4,7,8,9,11,12,14,15,16,17-dodecahydro-1H-cyclopenta[a]phenanthren-3-ol	[31]
80		(3S,5S)-17-(5-ethyl-6-methylheptan-2-yl)-10,13-dimethyl-2,3,4,5,6,9,11,12,14,15,16,17-dodecahydro-1H-cyclopenta[a]phenanthren-3-ol	[31]
		(E)-3-(4-hydroxyphenyl)prop-2-enal	
		(1S,2R)-1-(4-hydroxy-3-methoxyphenyl)-2-[4-[[(3R,4R,5S)-5-(4-hydroxy-3-methoxyphenyl)-4-(hydroxymethyl)oxolan-3-yl]methyl]-2-methoxyphenoxy]propane-1,3-diol	
81	**Acyclic peptides**	MCo-3	[24]
82		MCo-4	[24]
83		MCo-5	[24]
84		MCo-6	[24]
85		MCoCC-1	[22]
86		MCoCC-2	[22]
87		MCoTI-III	[23]
88		MCoTI-V	[28]
89		MCoTI-I	[23]
90	Cyclic peptides	(3S,6S,9S,12S,15S)-3,9-bis[(2S)-butan-2-yl]-6,12,15-tris(2-methylpropyl)-1,4,7,10,13-pentazacyclopentadecane-2,5,8,11,14-pentone	[32]
91		Clavatustide c	[32]
92		MCoTI-II	[23]
92	Trypsin inhibitors	MCoTI-II	
89		MCoTI-I	
93		MCoCI	[27]
87		MCoTI-III	
94		MCCTI-1	[21]
95	Proteins	Cochinin B	[25]
96		Cochinchinin	[26]
97		(2E,4E)-nona-2,4-dienal	[30]
98		Heptacosane	[34]
99		Pentatriacontan-18-one	[34]
100	Others	[(2S)-2-[(2R)-4-hexadecanoyloxy-3-hydroxy-5-oxo-2H-furan-2-yl]-2-hydroxyethyl] hexadecanoate	[30]
101		(E)-3-(4-hydroxyphenyl)prop-2-enal	[32]
102		4-hydroxybenzoic acid	[32]
103		5,5-dimethyl-cyclohex-3-en-1-ol	[30]

**Table 2 molecules-24-03949-t002:** Anti-tumor effects of Momordicae Semen extracts.

Solvent	Tumor	Cancer Cell	Effect	Observation	Refs
Water	Lung cancer	A549	Inhibit proliferation and migration	In vitro	[37]
	Breast cancer	ZR-75-30	Inhibit proliferation and migration	In vitro	[38]
	Melanoma	MM418C1,D24	Lowered vitality	In vitro	[39]
	Human cervical epithelial carcinoma	Human cervical epithelial carcinoma	—	In vitro	[25]
Ethanol	Breast cancer	MDA-MB-231	Inducing G2/M block and apoptosis	In vitro	[40]
	Lung cancer	A549, H1299	Inducing apoptosis, inhibiting invasion and migration	In vitro	[41]
	Gastric cancer	SGC7901, MKN-28	Apoptosis-inducing	In vitro	[42]
	Esophagus cancer	TE-13, Kyse30	Rhoa-MAPK signaling pathway is regulated	In vitro	[43]
	Human cervical epithelial carcinoma	Human cervical epithelial carcinoma	—	In vitro	[44]
Ethyl acetate	Breast cancer	MDA-MB-231	G2 cell cycle arrest, apoptosis and growth inhibition	In vitro and in vivo	[30]
	Melanoma	B16-F1	Inhibit proliferation and induce differentiation	In vitro and in vivo	[45]

**Table 3 molecules-24-03949-t003:** The general pharmacological effects and mechanisms of Momordicae Semen.

Pharmacological Effects	Extracts or Compounds	Mechanism	Refs
**Anti-inflammatory**	Saponins	IL-6↓, NO↓, IκBα↓, Src, Syk↓	[29,63]
**Antioxidant**	Chymotrypsin inhibitor	Reactive oxygen species↓	[64]
**Immune adjuvant**	Ethanol extract	IgG titers↑	[65]
**Antibacterial**	Water extract, saponins	Inhibit growth	[66]
**Kidney cell protecting**	Ethanol extract	Blocking the MAPKs signaling cascade	[67]
**Ulcer healing**	Ethanol extract	Mucus synthesis↑, cPLA_2_, 5-LOX↓ Pro-inflammatory cytokines↓	[68]
**Induce neurite outgrowth**	17-kda protein	—	[69]
**Reproductive protecting**	Aril extract	—	[70]

Abbrevation: NO: Nitric oxide; IL-6: interleukin- 6; IκBα: inhibitory subunit of NF-kB; Src: Scr kinase; Syk: Sky kinase; IgG: serum immunoglobulin G; MAPK: Mitogen-activated protein kinase; cPLA_2_: cytosolic phospholipaseA_2_; 5-LOX: 5-lipoxygenase.

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
