# Peer review of "A Potential Anti-Tumor Herb Bred in a Tropical Fruit: Insight into the Chemical Components and Pharmacological Effects of Momordicae Semen"

_molecules, 2019, doi:10.3390/molecules24213949_

Round 1

Reviewer 1 Report

Specific comments to the authors

The authors of the re-submitted review with the title “A Potential Anti-Tumor Herb Bred in a Tropical Fruit: Insight into the Chemical Components and Pharmacological Effects of Momordicae Semen” present an very intensive and re-structured overview of chemical compounds of Momordicae semen in the relation to possible mechanistic pathways, therapeutic options in human diseases (from chronic and degenerative diseases to different cancer entities) and toxicity profiling based on in-vitro and in-vivo-experiments.

Overall, the manuscript gives some interesting clinic-pharmacological aspects of different chemical solvent extracts of Momordica cochinchinensis seed in different clinical relevant chronic and degenerative diseases as well as human cancer entities. Although most of the previous concerns of the first submission (see submission of manuscript “molecules-550897”) were adequately addressed by the authors in the re-submitted version, some minor corrections should be performed by the authors as mentioned below:

# Abstract: The used abbreviation MS of “Momordicae Semen” is not very clever, since this is the standard abbreviation of “Multiple sclerosis”. Please change adequately, sorry.

# Discussion: The authors should shortly discuss the specific role of solvents in the chemical purification of Momordicae Semen in the detecting of the clinico-therapeutical relevant chemical substances in future.

Author Response

Response to Reviewer 1 Comments

Point 1: # Abstract: The used abbreviation MS of “Momordicae Semen” is not very clever, since this is the standard abbreviation of “Multiple sclerosis”. Please change adequately, sorry.

Response 1: We are very grateful for your suggestion. We have changed the abbreviation of “Momordicae Semen” for SEM (line 19).

Point 2: # Discussion: The authors should shortly discuss the specific role of solvents in the chemical purification of Momordicae Semen in the detecting of the clinico-therapeutical relevant chemical substances in future.

Response 2: Thank you for your suggestions. We have discussed the specific role of solvents in the chemical purification of Momordicae Semen in the detecting of the clinico-therapeutical relevant chemical substances in future (line 404-412).

Reviewer 2 Report

Molecules (molecules-621762), Comments to the Authors:

Title: A Potential Anti-Tumor Herb Bred in a Tropical Fruit: Insight into the Chemical Components and Pharmacological Effects of Momordicae Semen

Comments

The submitted review discussed phytochemical constituents and pharmacological activities of Gac fruit (Momordica cochinchinensis Spreng). The chemical composition includes saponins, fatty acids, volatile constituents, proteins and peptides and other components. The effect and mechanism of Gac and its extract on breast cancer, gastric cancer, lung cancer, esophagus cancer, melanomas and human cervical epithelial carcinoma were discussed. In addition, its anti-oxidant, anti-inflammatory and other pharmacological effects were also analyzed.

I think the review can be accepted for publication after the authors respond to the following comments:

The authors should provide a photo for the fruit. Compounds numbers in the text should match the figures. The quality of the drawing is poor. The authors should redraw the chemical structures using a professional software. The authors should add a diagram summarizing the anti-melanoma effects of the seeds. The authors should add a conclusion. There several reviews on the phytochemical constituents of Gac including: a) Journal of Functional Foods, Volume 62, November 2019, Gac (Momordica cochinchinensis Spreng) fruit: A functional food and medicinal resource (Review) Do, T.V.T., Fan, L., Suhartini, W., Girmatsion, M.; b) International Journal of Food Science and Technology, Volume 50, Issue 3, 1 March 2015, Pages 567-577, Gac fruit (Momordica cochinchinensis Spreng.): A rich source of bioactive compounds and its potential health benefits (Review), Chuyen, H.V., Nguyen, M.H., Roach, P.D., Golding, J.B., Parks, S.E.; c) Food Reviews International, Volume 29, Issue 1, January 2013, Pages 92-106, Gac Fruit: Nutrient and Phytochemical Composition, and Options for Processing (Review) Kha, T.C., Nguyen, M.H., Roach, P.D., Parks, S.E., Stathopoulos, C. The authors should highlight the importance of their review compared to previous reviews. They should also cite these reviews in their submitted review.

Author Response

Response to Reviewer 2 Comments

Point 1: The authors should provide a photo for the fruit.

Response 1: We are very grateful for your suggestion. The photo of the fruit was showed in this manuscript (figure 1)

Point 2: Compounds numbers in the text should match the figures.

Response 2: Thank you for your suggestions. We have matched the figures to compounds numbers. And the number of the compound in Table 1 corresponds to the number of the compound in the structure picture(figure2-8).

Point 3: The quality of the drawing is poor. The authors should redraw the chemical structures using a professional software.

Response 3: We are very sorry for our negligence and the chemical structures are redrawn(figure2-8).

Point 4: The authors should add a diagram summarizing the anti-melanoma effects of the seeds. The authors should add a conclusion. 

Response 4: Thank you for your suggestions. We have added a diagram summarizing the anti-melanoma effects of the seeds in figure 10. And conclusion was also added(line 218-222).

Point 5: There several reviews on the phytochemical constituents of gac including: a) Journal of Functional Foods, Volume 62, November 2019, Gac (Momordica cochinchinensis Spreng) fruit: A functional food and medicinal resource (Review) Do, T.V.T., Fan, L., Suhartini, W., Girmatsion, M.; b) International Journal of Food Science and Technology, Volume 50, Issue 3, 1 March 2015, Pages 567-577, Gac fruit (Momordica cochinchinensis Spreng.): A rich source of bioactive compounds and its potential health benefits (Review), Chuyen, H.V., Nguyen, M.H., Roach, P.D., Golding, J.B., Parks, S.E.; c) Food Reviews International, Volume 29, Issue 1, January 2013, Pages 92-106, Gac Fruit: Nutrient and Phytochemical Composition, and Options for Processing (Review) Kha, T.C., Nguyen, M.H., Roach, P.D., Parks, S.E., Stathopoulos, C. The authors should highlight the importance of their review compared to previous reviews. They should also cite these reviews in their submitted review.

Response 5: Thank you for your suggestions. We have cited these review (line 70,305,366,)  and the previous reviews were compared as follows:

(a)Gac (Momordica cochinchinensis Spreng) fruit: A functional food and medicinal resource. 

Firstly, research purposes are different. The purpose of paper a is to summarize gac’s cultivation reports, potential bioactive compounds, processing, uses, and storage of whole fruit as well as its products. However, our manuscript will provide new ideas for the treatment of cancer and other diseases, and a reference for the further research into complementary and alternative medicine. Secondly, the objects of concern are different. Paper a focused on gac fruit but our manuscript mainly focused on the seeds bred in gac fruit. Thirdly, paper a describes the pharmacological activity of the fruit, and the description of the activity against the tumor is small. However, this manuscript describes the pharmacological activity of seeds and mainly focused on its anti-tumor potential. Besides, the description of anti-tumor activity and chemical composition is more abundant.

 (b) Gac fruit (Momordica cochinchinensis Spreng.): A rich source of bioactive compounds and its potential health benefits.

Firstly, research purposes are different. The paper b aims to review on the nutritional composition, biological activities and processing of gac fruit. However, our manuscript will provide new ideas for the treatment of cancer and other diseases, and a reference for the further research into complementary and alternative medicine. Secondly, the objects of concern are different. Paper b focused on gac fruit but this manuscript mainly focused on the seeds bred in gac fruit. Thirdly, in paper b, the biological activity of the fruit is described and is not detailed enough and comprehensive. On the contrary, our manuscript describes the pharmacological activity of the seeds and is more comprehensive. In addition, except antioxidant, anti-inflammatory and other activities, our manuscript also mentions pharmacological effects such as induction of ganglia and reproductive protection.

(C) Gac Fruit: Nutrient and Phytochemical Composition, and Options for Processing.

Firstly, research purposes are different. The paper c aims to review the traditional uses and production of gac fruit, fruit nutrient and phytochemical composition, and the use of gac products as nutrient supplements and natural food colorants. However, our manuscript will provide new ideas for the treatment of cancer and other diseases, and a reference for the further research into complementary and alternative medicine. Secondly, the objects of concern are different. Paper a focused on gac fruit but our manuscript mainly focused on the seeds bred in gac fruit. Thirdly, paper c suggests that gac fruit products will have the potential to be widely used in beverages such as glutinous rice, yoghurt, pasta and other foods. Instead, this manuscript indicated the potential of seeds to be used in clinical. Fourthly, bioactive components of the fruit are concentrated in carotenoids with nutritional value in paper c but the compounds in seed are relatively comprehensive listed in our manuscript. Fifthly, biological activity is not mentioned in paper c. On the contrary, our manuscript details the biological activity of the seed and the pharmacological activities of different solvent extracts were compared.

In addition, Cytotoxic and anti-inflammatory constituents from Momordica cochinchinensis seeds (Fitoterapia. 2019 Oct 17:104360. doi: 10.1016/j.fitote.2019.104360. [Epub ahead of print]) was also added in our manuscript.

Reviewer 3 Report

The paper describes a review about chemical composition and biological activities of Gac fruit.   Some corrections should be done before the publication of the paper, such as:

Figure 1: structure 10 was represented wrong.

Figure 2: components 19-21 are not fatty acids. The figure must be revised. 

Figure 3: component 62 was detected after a derivatization process. The structure with silanyl groups is not a natural product.  Therefore the correct structure should be shown.

In table 1, the column "Kind" should be renamed as chemical class.

In table 1, triterpenoids and phenolic acids are included in fatty acids section. These data should be corrected.

In table 1, compounds listed in the section "others" could be splitted in different chemical classes such as, steroids, triterpenes, lignoids, cyclic peptides, etc..... The table should be reorganized and must be more informative to the readers.

Table 1: Several names of compounds are incorrect and not in accordance to IUPAC nomenclature rules. Examples: Hexadec-11-enoic acid instead 11-hexadecenoic acid;   Heptan-2-ol instead 2-heptanol.  The names of compounds from Table 1 must be checked.

A previous review about the nutritional composition, biological activities and processing of Gac fruit was published : “Chuyen, H. V., Nguyen, M. H., Roach, P. D., Golding, J. B. and Parks, S. E. (2015), Gac fruit (Momordica cochinchinensis Spreng.): a rich source of bioactive compounds and its potential health benefits. Int J Food Sci Technol, 50: 567-577. doi:10.1111/ijfs.12721”. Why the authors do not mentioned this paper in the references ?   Additionally, should be done a comparison between the published review and the present paper.

Author Response

Response to Reviewer 3 Comments

Point 1: Figure 1: structure 10 was represented wrong.

Response 1: We are very sorry for our negligence. We have corrected structure 10(figure 3).

Point 2: Figure 2: components 19-21 are not fatty acids. The figure must be revised.

Response 2: We are very sorry for our negligence. We have moved components 19-21 from fatty acids.

Point 3: Figure 3: component 62 was detected after a derivatization process. The structure with silanyl groups is not a natural product.  Therefore the correct structure should be shown.

Response 3: We are very sorry for our negligence. In the relevant literature(GC-MS Analysis of Chemical Constituents of Volatile Oil from Mubiezi), we learned that compound No. 62 is a chemical component in seeds. In the literature, steam distillation is used and the structure of the possible compounds may change. We did not find the original form of this compound. in order to avoid misunderstanding,we removed this compound.

Point 4: In table 1, the column "Kind" should be renamed as chemical class.

Response 4: Thank you for your suggestions. We have renamed "Kind" as chemical class.

Point 5: In table 1, triterpenoids and phenolic acids are included in fatty acids section. These data should be corrected.

Response 5: We are very sorry for our negligence. We have corrected fatty acids section and removed  triterpenoids and phenolic acids .

Point 6: In table 1, compounds listed in the section could be splitted in different chemical classes such as, steroids, triterpenes, lignoids, cyclic peptides, etc..... The table should be reorganized and must be more informative to the readers.

Response 6: Thank you for your suggestions. We have reorganized table1 and splitted "others" in triterpenes, lignoids, steroids, acyclic peptide ,cyclic peptides, protein, trypsin inhibitor.

Point 7: Table 1: Several names of compounds are incorrect and not in accordance to IUPAC nomenclature rules. Examples: Hexadec-11-enoic acid instead 11-hexadecenoic acid;   Heptan-2-ol instead 2-heptanol.  The names of compounds from Table 1 must be checked.

Response 7: We are very sorry for our negligence. We have changed the names of compounds to IUPAC names. And the component Clavatustide c(91, table1) and newly added Lignin (72-76, table 1) are from Cytotoxic and anti-inflammatory constituents from Momordica cochinchinensis seeds (Fitoterapia. 2019 Oct 17:104360. doi: 10.1016/j.fitote.2019.104360. [Epub ahead of print]). However, we do not find their IUPAC names. And we cite the names defined by the author of the paper.

Point 8: A previous review about the nutritional composition, biological activities and processing of Gac fruit was published : “Chuyen, H. V., Nguyen, M. H., Roach, P. D., Golding, J. B. and Parks, S. E. (2015), Gac fruit (Momordica cochinchinensis Spreng.): a rich source of bioactive compounds and its potential health benefits. Int J Food Sci Technol, 50: 567-577. doi:10.1111/ijfs.12721”. Why the authors do not mentioned this paper in the references ? Additionally, should be done a comparison between the published review and the present paper.

Response 8: Thank you for your suggestions. We compared to previous reviews.

The differences are listed as follows: first, this manuscript focuses on the seeds bred in gac fruit but that paper focused on the nutritional composition, biological activities and processing of gac fruit. Second, we focus on the chemical composition and pharmacological activity of the seeds and paid attention to the anti-tumor effect but that paper mainly concerned with the pharmacological activity of the fruit. Thirdly, the structure of compounds and names are showed  in this manuscript. 

(a)Gac (Momordica cochinchinensis Spreng) fruit: A functional food and medicinal resource. 

Firstly, research purposes are different. The purpose of paper a is to summarize gac’s cultivation reports, potential bioactive compounds, processing, uses, and storage of whole fruit as well as its products. However, our manuscript will provide new ideas for the treatment of cancer and other diseases, and a reference for the further research into complementary and alternative medicine. Secondly, the objects of concern are different. Paper a focused on gac fruit but our manuscript mainly focused on the seeds bred in gac fruit. Thirdly, paper a describes the pharmacological activity of the fruit, and the description of the activity against the tumor is small. However, this manuscript describes the pharmacological activity of seeds and mainly focused on its anti-tumor potential. Besides, the description of anti-tumor activity and chemical composition is more abundant.

 (b) Gac fruit (Momordica cochinchinensis Spreng.): A rich source of bioactive compounds and its potential health benefits.

Firstly, research purposes are different. The paper b aims to review on the nutritional composition, biological activities and processing of gac fruit. However, our manuscript will provide new ideas for the treatment of cancer and other diseases, and a reference for the further research into complementary and alternative medicine. Secondly, the objects of concern are different. Paper b focused on gac fruit but this manuscript mainly focused on the seeds bred in gac fruit. Thirdly, in paper b, the biological activity of the fruit is described and is not detailed enough and comprehensive. On the contrary, our manuscript describes the pharmacological activity of the seeds and is more comprehensive. In addition, except antioxidant, anti-inflammatory and other activities, our manuscript also mentions pharmacological effects such as induction of ganglia and reproductive protection.

(C) Gac Fruit: Nutrient and Phytochemical Composition, and Options for Processing.

Firstly, research purposes are different. The paper c aims to review the traditional uses and production of gac fruit, fruit nutrient and phytochemical composition, and the use of gac products as nutrient supplements and natural food colorants. However, our manuscript will provide new ideas for the treatment of cancer and other diseases, and a reference for the further research into complementary and alternative medicine. Secondly, the objects of concern are different. Paper a focused on gac fruit but our manuscript mainly focused on the seeds bred in gac fruit. Thirdly, paper c suggests that gac fruit products will have the potential to be widely used in beverages such as glutinous rice, yoghurt, pasta and other foods. Instead, this manuscript indicated the potential of seeds to be used in clinical. Fourthly, bioactive components of the fruit are concentrated in carotenoids with nutritional value in paper c but the compounds in seed are relatively comprehensive listed in our manuscript. Fifthly, biological activity is not mentioned in paper c. On the contrary, our manuscript details the biological activity of the seed and the pharmacological activities of different solvent extracts were compared.

In addition, Cytotoxic and anti-inflammatory constituents from Momordica cochinchinensis seeds (Fitoterapia. 2019 Oct 17:104360. doi: 10.1016/j.fitote.2019.104360. [Epub ahead of print]) was also added in our manuscript.

Round 2

Reviewer 2 Report

Molecules (molecules-621762 - Revised Version), Comments to the Authors:

Title: A Potential Anti-Tumor Herb Bred in a Tropical Fruit: Insight into the Chemical Components and Pharmacological Effects of Momordicae Semen

Comments

After reading the authors response to my comments, I believe all my remarks were answered and I think the review can be accepted for publication.

Reviewer 3 Report

The paper describes a review about chemical composition and biological activities of Gac fruit.   The suggestions were implemented in the text and the doubts removed.

I recommend the publication of the review.